# HMG-CoA reductase inhibitors and COVID-19 mortality in Stockholm, Sweden: A registry-based cohort study

**Rita Bergqvist**[1☯]*, **Viktor H. Ahlqvist**[1☯], **Michael Lundberg**[1,2], **Maria-Pia Hergens**[1,3], **Johan Sundström**[4,5], **Max Bell**[6,7], **Cecilia Magnusson**[1,2]

**1** Department of Global Public Health, Karolinska Institutet, Stockholm, Sweden, **2** Centre for Epidemiology and Community Medicine, Region Stockholm, Stockholm, Sweden, **3** Department of Communicable Disease Control and Prevention, Region Stockholm, Stockholm, Sweden, **4** Department of Medical Sciences, Uppsala University, Uppsala, Sweden, **5** The George Institute for Global Health, University of New South Wales, Sydney, Australia, **6** Department of Perioperative Medicine and Intensive Care, Karolinska University Hospital, Stockholm, Sweden, **7** Department of Physiology and Pharmacology, Karolinska Institutet, Stockholm, Sweden

☯ These authors contributed equally to this work.
* rita.bergqvist@stud.ki.se

**Data Availability Statement:** Swedish privacy law prohibits us from making register data publicly available. The data supporting our findings were used under license and ethical approval for the

## Abstract

### Background

The relationship between statin treatment and Coronavirus Disease 2019 (COVID-19) mortality has been discussed due to the pleiotropic effects of statins on coagulation and immune mechanisms. However, available observational studies are hampered by study design flaws, resulting in substantial heterogeneity and ambiguities. Here, we aim to determine the relationship between statin treatment and COVID-19 mortality.

### Methods and findings

This cohort study included all Stockholm residents aged 45 or older ($N$ = 963,876), followed up from 1 March 2020 until 11 November 2020. The exposure was statin treatment initiated before the COVID-19-pandemic, defined as recorded statin dispensation in the Swedish Prescribed Drug Register between 1 March 2019 and 29 February 2020. COVID-19-specific mortality was ascertained from the Swedish Cause of Death Registry. Hazard ratios (HRs) were calculated using multivariable Cox regression models. We further performed a target trial emulation restricted to initiators of statins.

In the cohort (51.6% female), 169,642 individuals (17.6%) were statin users. Statin users were older (71.0 versus 58.0 years), more likely to be male (53.3% versus 46.7%), more often diagnosed with comorbidities (for example, ischemic heart disease 23.3% versus 1.6%), more frequently on anticoagulant and antihypertensive treatments, less likely to have a university-level education (34.5% versus 45.4%), and more likely to have a low disposable income (20.6% versus 25.2%), but less likely to reside in crowded housing (6.1% versus 10.3%).

current study. Readers interested in obtaining microdata or replicating our study may seek similar approvals and inquire through Statistics Sweden. For further advice see: https://www.scb.se/en/services/guidance-for-researchers-and-universities/, or contact Statistics Sweden at: mikrodata@scb.se.

**Funding:** The author(s) received no specific funding for this work.

**Competing interests:** I have read the journal's policy and the authors of this manuscript have the following competing interests: JS has stock ownership in companies providing services to Itrim, Amgen, Janssen, Novo Nordisk, Eli Lilly, Boehringer, Bayer, Pfizer and AstraZeneca, outside the submitted work.

**Abbreviations:** ACE, angiotensin-converting enzyme; ATC, anatomical therapeutic classification; BMI, body mass index; COVID-19, Coronavirus Disease 2019; HR, hazard ratio; ICD, International Classification of Diseases; ICU, intensive care unit; LISA, longitudinal integrated database for health insurance and labor market studies; OR, odds ratio; TIA, transient ischemic attack.

A total of 2,545 individuals died from COVID-19 during follow-up, including 765 (0.5%) of the statin users and 1,780 (0.2%) of the nonusers. Statin treatment was associated with a lowered COVID-19 mortality (adjusted HR, 0.88; 95% CI, 0.79 to 0.97, $P = 0.01$), and this association did not vary appreciably across age groups, sexes, or COVID-19 risk groups. The confounder adjusted HR for statin treatment initiators was 0.78 (95% CI, 0.59 to 1.05, $P = 0.10$) in the emulated target trial. Limitations of this study include the observational design, reliance on dispensation data, and the inability to study specific drug regimens.

## Conclusions

Statin treatment had a modest negative association with COVID-19 mortality. While this finding needs confirmation from randomized clinical trials, it supports the continued use of statin treatment for medical prevention according to current recommendations also during the COVID-19 pandemic.

---

## Author summary

### Why was this study done?

- Lipid-lowering HMG-CoA reductase inhibitors (statins) may influence Coronavirus Disease 2019 (COVID-19) mortality via their pleiotropic effects on coagulation and the immune system.

- Previous studies on this topic have been inconsistent, so we performed a population-based cohort study to investigate the relationship between statins and COVID-19 mortality.

### What did the researchers do and find?

- Using data from Swedish health registers, a cohort of 963,876 residents of Stockholm, Sweden, were followed from 1 March 2020 until 11 November 2020.

- Prescription dispensation data were matched to healthcare data and the Swedish Cause of Death Registry and analyzed using multivariable cause-specific survival analysis.

- Statin treatment was associated with a moderately lower risk of COVID-19 mortality (hazard ratio (HR), 0.88; 95% CI, 0.79 to 0.97), after accounting for a series of preexisting health conditions and other factors. The association was corroborated by sensitivity analyses and did not vary substantially across risk groups.

### What do these findings mean?

- These findings suggest that statin treatment may have a modest preventive therapeutic effect on COVID-19 mortality, although randomized trials are needed to determine the causality of the observed association.

- In summary, the findings support the continued use of statins for conditions such as cardiovascular disease and hyperlipidemia, in line with current recommendations, during the COVID-19 pandemic.

## Introduction

The Coronavirus Disease 2019 (COVID-19) pandemic continues to evolve. Vaccines are underway, and possible treatments are being tested in trials and clinics. The understanding of the pathophysiology of the disease continues to grow along with the knowledge of risk factors for severe COVID-19 disease and death. Hyperinflammation and hypercoagulability have been identified as central to the development of severe COVID-19 disease and complications [1–4]. Hence, drugs that modulate the host immune response and inhibit thrombosis and vascular dysfunction have received widespread attention. Special attention has been given to low-risk and low-cost treatments that can be easily administered across different settings [5–15].

Lipid-lowering HMG-CoA reductase inhibitors (statins), which are used in vast groups of patients for the prevention and treatment of cardiovascular diseases, are one of the pharmaceutical classes that have been proposed as possible preventive and/or adjuvant treatment of severe COVID-19. In brief, statins inhibit the enzyme HMG-CoA reductase central to the endogenous synthesis of cholesterol. This inhibition results in decreased levels of harmful LDL cholesterol in blood plasma. Statins are also known to have pleiotropic effects, including immune modulation, decreased inflammation in blood vessels, improved endothelial function, decreased thrombocyte aggregation, decreased risk of thrombosis, increased fibrinolysis, and stabilization of atherosclerotic plaques [16–18].

Several observational studies have investigated statins and report, with some discrepancies, an association between statin use and decreased COVID-19 mortality [19–27]. However, many previous studies have failed to avoid the potential pitfalls of observational studies, like those commonly described under the emulated trial framework [28]. In addition, most previous studies have had small sample sizes, owing to the fact that they have been based on early pandemic cases, and often included only hospitalized populations. As such, a recently published meta-analysis showed substantial heterogeneity ($i^2$ 90%) between previous studies, likely due to methodological discrepancies [29]. For example, certain studies of statins and COVID-19 mortality have suggested a substantial excessive risk (odds ratio (OR) 6.21), while others have suggested implausible benefits (OR 0.46) [29].

Pending ongoing clinical trials [30–38], we present a large population-based observational study using Swedish health register data, with the aim of determining the relationship between statin treatment and COVID-19 mortality, scrutinize if this relationship is robust to target trial emulation of statin initiation and examine whether such a relationship is consistent across age groups, sexes, and COVID-19 risk groups.

## Methods

### Study design and data sources

We designed a total population study, covering all Stockholm residents meeting the study criteria (see below), based on routinely collected data from the following registries: the longitudinal integrated database for health insurance and labor market studies (LISA) [39] for information on education and income, the VAL databases of Region Stockholm containing information from both hospitals and outpatient clinics on preexisting conditions according to

the International Classification of Diseases (ICD), as well as data from the Swedish Prescribed Drug Register [40] on collected prescriptions coded by anatomical therapeutic classification (ATC) codes. Data on COVID-19 related deaths were collected from the Swedish Cause of Death Registry [41]. Registries and records were linked through the personal identity number assigned to each Swedish resident [42]. The study has been approved by the Regional Ethical Review Board, Stockholm (2021–00810). This study is reported as per the Strengthening the Reporting of Observational Studies in Epidemiology (STROBE) guideline (S1 STROBE Checklist).

## Study population

The study included all individuals aged 45 or older, residing in Stockholm county in 2019 as well as 1 March 2020, as recorded in the Total Population Registry. We chose to only study those older than 45 to effectively exclude >99% of pregnant women, as statins are contraindicated during pregnancy. Furthermore, the lower age limit enhances the stability of our covariate control, as there had been few COVID-19 deaths among young people during the study period. Furthermore, statins are contraindicated in active liver disease, individuals diagnosed with liver disease (ICD-10: K70 to K77) up to 5 years prior to the start of the study (i.e., 1 Jan 2015 or later) were hence excluded (*n* = 2,103). In total, 963,876 individuals were retained for analysis.

## Exposure(s)

The primary exposure was defined as one or more collected prescriptions of any type of statin (ATC code C10AA), between 1 March 2019 and 29 February 2020 (allowing for both possible intermediate- and short-term pharmacological effects). We restricted exposure collection to the period before the outbreak of the pandemic to avoid possible bias originating from altered healthcare seeking and drug collecting behaviors during the outbreak.

## Follow-up/outcomes

Individuals were followed until death from COVID-19, other deaths, or 11 November 2020, whichever occurred first. The primary outcome, death from COVID-19, was established using the Swedish Cause of Death Registry, which is based on the death certificate filled in by physicians. This certification is based on a clinical evaluation supported by, for example, radiology, microbiological analysis, and/or assessment of symptoms. For the main analysis, all deaths for which COVID-19 was registered as the main cause or a contributing cause were included.

## Statistical analysis

Using measures of central tendency and dispersion, we descriptively presented our sample by statin exposure. To quantify the relationship between statin treatment and COVID-19-specific mortality, we employed cause-specific Cox regression, using days since 1 March 2020 as the underlying timescale.

We present unadjusted and adjusted hazard ratios (HRs) and their 95% confidence intervals, adjusting for sex (categorical), age (continuous), country of birth (categorical), the highest level of obtained education (categorical), annual disposable income in 2018 (quartiles), residential area (categorical), household crowding (m$^2$ per person) (deciles), nursing home (yes/no), exposure to angiotensin-converting enzyme (ACE) inhibitors (yes/no), angiotensin receptor blockers (yes/no), and anticoagulants (yes/no) as well as a broad range of conditions (yes/

no); atrial fibrillation, cancer, cerebrovascular disorders, chronic kidney disease, chronic lower respiratory disease, dementia (inc. Alzheimer disease), type 1 and type 2 diabetes, dyslipidemia, heart failure, hypertension, ischemic heart disease, obesity, other neurological conditions, peripheral vascular disease, pure hypercholesterolemia, renal failure stage 3 to 5, stroke and transient ischemic attack (TIA), and vascular disease (see S1 Table for ICD-10/ATC codes).

We evaluated the proportional hazard assumption by testing the slope of the Martingale residuals as well as through visual inspection of log of minus log plots, finding limited evidence of a violation of the proportional hazard assumption. As the fraction of residents with missing covariate information was small (6.2%), all analyses were performed using the so-called missing indicator approach, where residents with missing covariate information are treated as a distinct category in each covariate. We repeated our main analysis using complete case, to examine a possible difference compared to the missing indicator approach.

Subgroup analyses were performed for age groups, sexes, different statin indications as well as COVID-19 risk groups (identified by authorities and/or in previous studies on risk factors [43]), and evaluated for heterogeneity using the Cochran's Q statistic. Computations were performed using SAS version 9.4. $P$ values <0.05 were considered statistically significant.

The rapid development of the COVID-19 pandemic prevented us from preregistering our statistical analysis plan, as we prioritized a rapid execution of our study.

### Sensitivity analyses

We performed several sensitivity analyses to scrutinize our results: We (1) restricted our main analysis to statin initiators (emulating a target trial with an intention-to-treat analysis [44]); (2) repeated the main analysis with COVID-19 as the main underlying cause of death as the outcome measure; (3) performed analysis of all-cause mortality without COVID-19 (positive outcome analysis [45]); and (4) performed our main analysis using the Fine-Gray method for competing risks [46], treating non-COVID deaths as competing events.

## Results

### Sample characteristics

A total of 963,876 residents aged 45 or older were included in our main analysis, out of which 496,865 individuals (51.6%) were female (Table 1). Among the residents, 169,642 individuals (17.6%) had collected at least one statin prescription during the year preceding the pandemic.

Statin users were older (median age of 71.0 years, compared to 58.0 years for nonusers) and more likely to be male (53.3% versus 46.7%). They had more often been diagnosed with comorbidities (ischemic heart disease 23.3% versus 1.6%, heart failure 9.45% versus 2.13%, hypertension 74.3% versus 24.7%, type 2 diabetes 32.6% versus 4.75%) and were more frequently on anticoagulant and antihypertensive treatments (ACE inhibitors 2.7% versus 1.1%; angiotensin receptor blockers 3.6% versus 1.6% and anticoagulants 5.0% versus 2.1%) compared to nonusers. A smaller share of statin users had university-level education (34.5% versus 45.4%), and statin users had a lower disposable income (Q1: 28.9% versus 23.4%, Q4: 20.6% versus 25.2%) but less crowded housing (decile 1: 6.1% versus 10.3%). There were no major differences between users and nonusers regarding country of origin or residential area.

### Main analysis of COVID-19 mortality

A total of 2,545 individuals died from COVID-19 during follow-up, including 765 (0.5%) of the statin users and 1,780 (0.2%) of the nonusers (unadjusted HR, 2.02; 95% CI, 1.85 to 2.20) (Fig 1). However, when adjusting for confounders, statin treatment was associated with a

**Table 1. Selected characteristics of the studied cohort.**

| Characteristic | Statin users (N = 169,642) | Nonusers (N = 794,234) |
|---|---|---|
| Sex, No. (%) | | |
| Female | 73,683 (43.4) | 423,182 (53.3) |
| **Age** | | |
| Years, Median (IQR) | 71.0 (64–78) | 58.0 (49–67) |
| 45–69 years, No. (%) | 75,801 (44.7) | 609,427 (76.7) |
| 70–79 years, No. (%) | 61,829 (36.5) | 122,806 (15.5) |
| ≥80 years, No. (%) | 32,012 (18.9) | 62,001 (7.8) |
| **Comorbidities/preexisting conditions, No. (%)** | | |
| Type 1 diabetes | 6,578 (3.9) | 4,481 (0.6) |
| Type 2 diabetes | 55,260 (32.6) | 37,721 (4.8) |
| Obesity | 15,793 (9.3) | 31,418 (4) |
| Dyslipidemia | 77,670 (45.8) | 23,205 (2.9) |
| Pure hypercholesterolemia | 3,507 (2.1) | 16,524 (2.1) |
| Dementia (inc. Alzheimer disease) | 5,480 (3.2) | 12,933 (1.6) |
| Hypertension | 125,978 (74.3) | 196,100 (24.7) |
| Heart failure | 16,030 (9.5) | 16,956 (2.1) |
| Vascular disease | 13,997 (8.3) | 19,789 (2.5) |
| Cerebrovascular disorders | 20,442 (12.1) | 12,593 (1.6) |
| Stroke and TIA | 6,741 (4) | 31,903 (4) |
| Chronic lower respiratory disease | 14,296 (8.4) | 27,848 (3.5) |
| Ischemic heart disease | 39,518 (23.3) | 12,599 (1.6) |
| Peripheral vascular disease | 4,776 (2.8) | 1,657 (0.2) |
| Renal failure stage 3–5 | 2,957 (1.7) | 14,025 (1.8) |
| Atrial fibrillation | 23,296 (13.7) | 33,965 (4.3) |
| Chronic kidney disease | 13,152 (7.8) | 13,501 (1.7) |
| Cancer | 27,232 (16.1) | 71,283 (9) |
| Other neurological conditions | 8,666 (5.1) | 20,750 (2.6) |
| Any of the above conditions | 51,959 (30.6) | 71,734 (9) |
| **Other medications, No. (%)** | | |
| ACE inhibitors | 4,524 (2.7) | 8,953 (1.1) |
| Angiotensin receptor blockers | 6,074 (3.6) | 12,642 (1.6) |
| Anticoagulants | 8,436 (5) | 16,362 (2.1) |
| **Socioeconomic status, No. (%)** | | |
| *Education* | | |
| Primary education | 36,602 (21.6) | 104,740 (13.2) |
| Secondary education | 69,759 (41.1) | 301,743 (38) |
| Tertiary education | 58,505 (34.5) | 360,418 (45.4) |
| Unknown | 4,776 (2.8) | 27,333 (3.4) |
| *Disposable income quartile* | | |
| Q1 | 49,102 (28.9) | 185,789 (23.4) |
| Q2 | 46,922 (27.7) | 187,751 (23.6) |
| Q3 | 35,868 (21.1) | 199,199 (25.1) |
| Q4 | 34,921 (20.6) | 200,039 (25.2) |
| Unknown | 2,829 (1.7) | 21,456 (2.7) |
| *Residing in a nursing home, No. (%)* | | |
| Yes | 3,473 (2.1) | 12,090 (1.5) |

(*Continued*)

**Table 1.** (Continued)

| Characteristic | Statin users (N = 169,642) | Nonusers (N = 794,234) |
|---|---|---|
| *Country of birth, No. (%)* | | |
| Sweden | 120,959 (71.3) | 570,543 (71.8) |
| *Household crowding (m² per person), No. (%)* | | |
| Decile 1 | 10,267 (6.1) | 82,114 (10.3) |
| Decile 2 | 10,026 (5.9) | 81,649 (10.3) |
| Decile 3 | 12,472 (7.4) | 80,133 (10.1) |
| Decile 4 | 14,231 (8.4) | 75,963 (9.6) |
| Decile 5 | 17,673 (10.4) | 76,617 (9.7) |
| Decile 6 | 17,975 (10.6) | 74,845 (9.4) |
| Decile 7 | 19,082 (11.3) | 73,227 (9.2) |
| Decile 8 | 20,883 (12.3) | 72,939 (9.2) |
| Decile 9 | 20,695 (12.2) | 70,777 (8.9) |
| Decile 10 | 21,149 (12.5) | 71,347 (9) |
| Unknown | 5,189 (3.1) | 34,623 (4.4) |

ACE, angiotensin-converting enzyme; IQR, interquartile range; TIA, transient ischemic attack.

moderately lower risk of COVID-19 mortality (HR, 0.88; 95% CI, 0.79 to 0.97). This association was similar across age groups (*P* = 0.821) and sexes (*P* = 0.657) and COVID-19 risk groups/indications for statin treatment (*P* = 0.727).

## Sensitivity analyses

In our emulated target trial (statin initiators only), the negative association between statin treatment and COVID-19 mortality remained but was not statistically significant (HR, 0.78; 95% CI, 0.59 to 1.05) (S1 Text). Restricting our analysis to those with COVID-19 as the main underlying cause of death did not alter our findings (S1 Text). Our positive outcome analysis was consistent with a protective effect of statins (as expected), although the estimated protective effect was greater than that of meta-analysis of randomized controlled trials examining statins for primary prevention in high-risk populations [47] (S1 Text). The fact that we were able to replicate the known association between statin treatment and reduced (overall) mortality gives further support for the validity of the results from our COVID-19 analysis. There was no difference between our main analysis and a complete case replication of our analysis (HR, 0.87; 95% CI, 0.79 to 0.97, *N* = 903,782). There was no difference between our main analysis and when treating non-COVID deaths as competing events using the Fine–Gray method (adjusted HR, 0.88; 95% CI, 0.80 to 0.98) (S2 Table).

## Discussion

### Interpretation of results

In this register-based cohort study of 963,876 Stockholm residents aged 45 or older, statin treatment was associated with a moderately lower risk of COVID-19 mortality. Adjusted HRs were largely consistent across age groups, sexes, as well as across different COVID-19 risk groups and indications for statin treatment. This result was corroborated by our sensitivity analysis explicitly emulating a target trial among initiators of statin treatment.

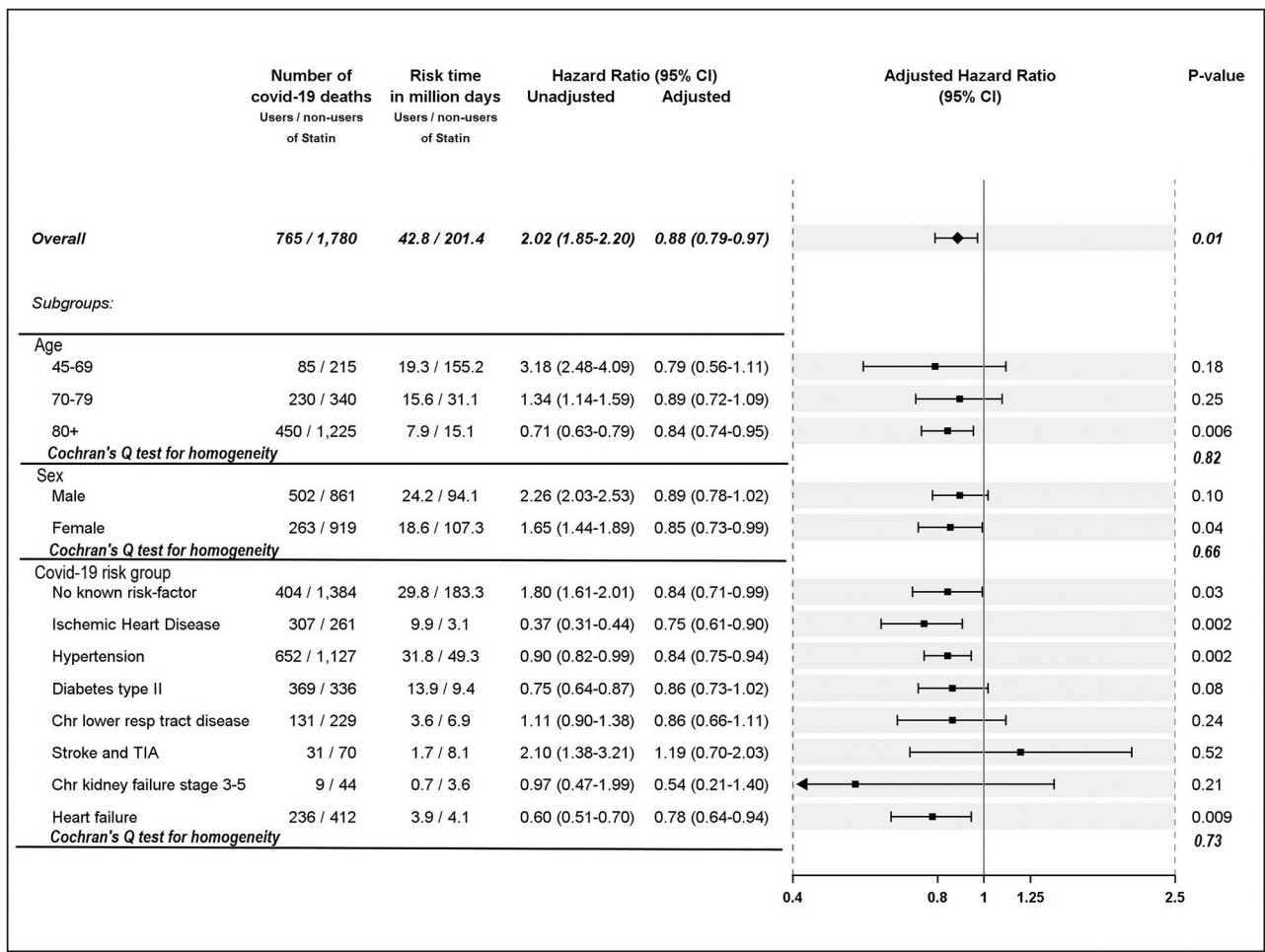

**Fig 1. HRs of death from COVID-19 in relation to dispensations of statins overall and the HRs of death from COVID-19 in relation to dispensations of statins within strata of age, sex, and within COVID-19 risk groups.** chr, chronic; CI, confidence interval; COVID-19, Coronavirus Disease 2019; HR, hazard ratio; TIA, transient ischemic attack.

Several cohort studies of hospitalized COVID-19 patients have reported negative associations between statin use and mortality [20–27], in line with our findings. These studies have been summarized in meta-analyses indicating an overall reduction of unfavorable COVID-19 outcomes (including death, intensive care unit (ICU) admission, and mechanical ventilation) of about 30% in statin users [48,49]. The meta-analyses also demonstrate substantial heterogeneity between studies (i² 90%) [29], and a recent large-scale study on severe COVID-19 and all-cause mortality in COVID-19–positive Danish citizens showed no association between statin use and either outcome [19].

Existing studies have been limited in their appreciation of the complexity of causal inference when using observational data and may thus be prone to the critical pitfalls of observational studies of clinical interventions. Such pitfalls have, for example, been clearly demonstrated in works on statins and cancer [28] and, indeed, in studies of COVID-19 [50]. These include failure to differentiate between new and prevalent users (leading to survival bias), using post-baseline information to establish exposure (leading to immortal time bias), introducing collider bias by sampling only hospitalized patients and/or conditioning on positive PCR test results or adjusting for potential mediators (such as disease severity).

## Strengths and limitations

Our study has several strengths. Most importantly, we applied a rigorous methodological approach to avoid the potential biases mentioned above, resulting in a study with high internal validity. To avoid conditioning on PCR testing, which may introduce collider bias (as previously observed in connection with COVID-19 research [50]), the outcome measure was based on data from The Swedish Cause of Death Register. This register contains all deaths determined by physicians to be caused (in full or in part) by COVID-19, based on testing or other diagnostics. Moreover, mortality carries a smaller risk of selection bias than outcome measures such as ICU admission or severity of disease.

Furthermore, we were able to control for a large set of important confounders using data collected in a standardized manner. We also avoid confounding that may arise from altered behavior during the COVID-19 pandemic. Specifically, the end of the period of exposure was set to 1 March 2020—the start of the pandemic in Stockholm—since the collection of prescriptions during a pandemic could be a proxy for certain behaviors affecting the risk of exposure to the virus (for example, the frequenting of public spaces). While we believe that this is a major strength of our study, such time restrictions imply that we include statin dispensations, which, in theory, could have been collected long before a COVID-19 infection. Indeed, individuals who collect a sole prescription in early 2019 will be classified as exposed. However, this is in line with the intention-to-treat principle. As such, our estimate may represent a conservative estimate of the causal effect of statins on COVID-19 mortality; our estimate can be interpreted as that obtained from an observational intention-to-treat analysis in the presence of noncompliance, which we believe is the clinically relevant causal contrast. Another important aspect is the differentiation in recommendations for different age groups during spring and summer; people aged 70 or older were recommended to isolate, which could render analyses based on dispensation during the spring and summer unrecoverable from such selection mechanisms.

Lastly, in context of other evidence on COVID-19, the duration of follow-up is long and the number of exposed cases high. Our study includes the total population of Stockholm County, whereas most previous studies were hospital-based. Hence, the results are likely to have higher external validity.

However, the study also has several important limitations. Firstly, as in any observational study, residual confounding cannot be ruled out. For example, we have not adjusted for smoking or body mass index (BMI), but only diagnosed obesity. Both smoking and high BMI are strongly linked to statin use [51], and high BMI has been identified as a risk factor for COVID-19 death [52]. Although we would expect this to result in an underestimation of a potential protective effect, we cannot dismiss the possibility that our findings may be explained by confounding.

Secondly, as in most pharmacoepidemiological studies, we have assumed that drug dispensation captures drug use correctly. This assumption is easily challenged since not all patients filling prescriptions adhere to treatment. This misclassification is, however, unlikely to explain our findings. Any nondifferential misclassification will generally dilute associations, and we judge a possible differential misclassification to bias our findings toward a positive association if adherence is particularly low in severely ill patients (including in COVID-19) or patients that have low health literacy and fewer means of avoiding the infection.

Lastly, we have not been able to study possible effects of dosage, type, or brand of statins. Specifically, despite our study being, the hereby largest study of statins and COVID-19 mortality, we had an insufficient sample size to perform such granular analysis. As such, our estimate should be interpreted as the mean of possibly heterogeneous effects.

Further aspects should be considered when interpreting our findings. This study does not concern the possible effect of statin treatment for treatment of severe COVID-19, but its possible preventive effect. Furthermore, although our duration of follow-up is longer than comparable studies, future studies could shed light on the long-term effects of statins on COVID-19 mortality and possible heterogeneity in effects over subsequent COVID-19 waves. Although we emulated a target trial with an intention-to-treat analysis, the study does not provide an analysis of the risk of adverse events in potential target groups. Safety evaluations are outside the scope of the current study as they require extensive clinical and qualitative safety data, and such evaluations should address several aspects of the overall concept of statin safety. Nonetheless, this has been addressed in previous clinical trials, but not in connection with this new potential indication (COVID-19 prevention). Although statins are considered to have a favorable safety profile [53] compared to many other drug classes discussed in connection with COVID-19, a clinical risk assessment must always be made. Finally, although our study has greater generalizability due to the general population design (as compared to hospital-based studies), its results should be interpreted in light of the Swedish COVID-19 strategy, which may have implications for the generalizability of our findings.

## Conclusions

The current pandemic constitutes a unique situation in that policy and treatment decisions are being continuously made on the basis of limited knowledge. In the absence of results from randomized clinical trials [30–38], our study can provide some guidance. Specifically, it gives further support for continuing statin treatment for conditions such as cardiovascular disease and hyperlipidemia, in line with current recommendations [54–57], during the COVID-19 pandemic.

## Supporting information

**S1 STROBE Checklist. Strengthening the Reporting of Observational Studies in Epidemiology (STROBE) checklist.**
(DOCX)

**S1 Table. Covariate definition according to ICD-10/ATC code and period of data collection.**
(DOCX)

**S2 Table. Sensitivity analysis using the Fine–Gray subdistribution hazard model.**
(DOCX)

**S1 Text. Extended details on sensitivity analyses.**
(DOCX)

## Author Contributions

**Conceptualization:** Rita Bergqvist, Viktor H. Ahlqvist, Michael Lundberg, Cecilia Magnusson.

**Data curation:** Michael Lundberg.

**Formal analysis:** Michael Lundberg.

**Methodology:** Rita Bergqvist, Viktor H. Ahlqvist, Michael Lundberg, Maria-Pia Hergens, Johan Sundström, Max Bell, Cecilia Magnusson.

**Project administration:** Cecilia Magnusson.

**Supervision:** Viktor H. Ahlqvist, Cecilia Magnusson.

**Visualization:** Michael Lundberg.

**Writing – original draft:** Rita Bergqvist, Viktor H. Ahlqvist.

**Writing – review & editing:** Michael Lundberg, Maria-Pia Hergens, Johan Sundström, Max Bell, Cecilia Magnusson.

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
