## [Editor Report · Decision Letter 0]

24 May 2021

Dear Dr Bergqvist, 

Thank you for submitting your manuscript entitled "HMG-CoA reductase inhibitor therapy and its relationship to COVID-19 mortality in the general population of Stockholm" for consideration by PLOS Medicine.

Your manuscript has now been evaluated by the PLOS Medicine editorial staff as well as by an academic editor with relevant expertise and I am writing to let you know that we would like to send your submission out for external peer review.

Please re-submit your manuscript within two working days, i.e. by May 26 2021 11:59PM.

Kind regards,

Callam Davidson

Associate Editor

PLOS Medicine

---

## [Decision Letter · Decision Letter 1]

28 Jun 2021

Dear Dr. Bergqvist,

Thank you very much for submitting your manuscript "HMG-CoA reductase inhibitor therapy and its relationship to COVID-19 mortality in the general population of Stockholm" (PMEDICINE-D-21-02232R1) for consideration at PLOS Medicine. 

Your paper was evaluated by an associate editor and discussed among all the editors here. It was also discussed with an academic editor with relevant expertise, and sent to independent reviewers, including a statistical reviewer. The reviews are appended at the bottom of this email and any accompanying reviewer attachments can be seen via the link below:

[LINK]

In light of these reviews, I am afraid that we will not be able to accept the manuscript for publication in the journal in its current form, but we would like to invite you to submit a revised version that addresses the reviewers' and editors' comments fully. You will appreciate that we cannot make a decision about publication until we have seen the revised manuscript and your response, and we expect to seek re-review by one or more of the reviewers. 

We hope to receive your revised manuscript by Jul 19 2021 11:59PM. Please email us (plosmedicine@plos.org) if you have any questions or concerns.

Please let me know if you have any questions, and we look forward to receiving your revised manuscript. 

Sincerely,

Callam Davidson, 

Associate Editor

PLOS Medicine

plosmedicine.org

Please update the title of the manuscript to match the style used by PLOS Medicine. “Prescriptions for HMG-CoA reductase inhibitors and COVID-19 mortality in Stockholm, Sweden: a registry-based cohort study”, or similar, would be suitable. 

Please confirm whether ‘Swedish secrecy law’ (in your response to the data availability question in the submission form) should be updated to ‘Swedish privacy law’.

Please structure the ‘Abstract’ using three sections: ‘Background’, ‘Methods and Findings’, and ‘Conclusions’. The ‘Objectives’ section in the current abstract can be rephrased as the final line of the ‘Background’ (“The objective of this study was to determine the relationship…” etc.). The ‘Methods’ and ‘Results’ sections can be combined, and the final sentence of the ‘Methods and Findings’ section ought to summarise 2-3 of the study’s main limitations (please begin this sentence “Limitations of this study include…”). 

Please present your conclusions in the past tense (e.g., “there was a modest negative association between statin treatment and COVID-19 mortality”).

Please state summary demographic information in the abstract and include relevant p-values. 

Please provide an ‘Author Summary’ section as outlined here: https://journals.plos.org/plosmedicine/s/revising-your-manuscript

Please remove the ‘Conflicts of interest’ and ‘Role of the funding source’ from the title page of the manuscript. If published, the relevant information will be included as metadata based on your responses to the submission forms.

Please update lines 191-192 to read “However, when adjusting for confounders, statin treatment was associated with a moderately lower risk of COVID-19 mortality (HR, 0.88; 95% CI, 0.79-0.97)”. Please similarly adjust other relevant statements throughout the manuscript (e.g. lines 221-222). 

If a protocol and/or statistical analysis plan were prepared in advance of this study, please could these be provided as supplementary files with the next revision. 

Please refer to any attached protocol document in the Methods section (main text) and highlight any analyses that were not prespecified.

Please make sure references do not contain any bold/italicized text and use “et al.” after listing the first six authors of a paper. 

In the reference list, please add “[preprint]” to all preprints, e.g., reference 26.

Thank you for providing a completed STROBE checklist. In the checklist, please refer to individual items by section (e.g., "Methods") and paragraph number, not by line or page numbers as these generally change in the event of publication.

Comments from the reviewers:

Reviewer #1: This is a well-executed observational study investigating the hypothesis that statins improve the COVID-related mortality. The extent of the study as well as the rigorous approach are wellcomed. 

Some of the limitations are well-explained. For example, the type of statin used was not explored, which is a missed opportunity seen the differences between the statines. Moreover, it is not explained why this sub-analysis was not done.

Also, the exposure was defined as one or more collected prescriptions: if only one prescription? Also, information about the continuation till "outcome" seems missing: what if in the last days-weeks, no statines were anymore given?

Although one could ask these and other questions, I still think this is a valuable study worth to be published, adding to the "pleiotropic" effects of statines.

Reviewer #2: The authors have done study statin in COVID-19. My comments are

1. Should be stated that chronic statin user before got COVID-19 affect the mortality. 

2. There was little data regarding how long did statin give protection. And this was retrospective cohort study, there was bias that could not be controlled. This issue should be added as the limitation of the study. 

3. Was there any that regarding the safety of statin users?

4. Different type of statin use may give different effect  should also be discussed

5. In this study was interesting, dementia, obesity and diabetes was higher in statin group. This issue should be discussed DOI: 10.1016/j.amjms.2020.10.026

6. The routine use of anticoagulant higher in statin group may give a confounder effect

7. How about the use of diabetes drugs? some give benefit https://doi.org/10.1007/s40200-021-00777-4

8. In table 1 should give p value to know if both group was same or differ in characteristics

9. 

Reviewer #3: My Review:

The authors present a large, Stockholm, Sweden population-based observational study of statin use and COVID-19 mortality. The study population included all residents of Stockholm over 45 yrs of age without liver disease. The exposure was any collection of a statin prescription from March 2019 through Feb 2020; outcome was death from COVID-19. They found that although unadjusted analysis revealed higher mortality for statin users (a sicker population), after extensive adjustment for confounders, statin use was associated with a reduction in COVID-19 mortality. Sensitivity analyses revealed this association to hold true for non-COVID deaths; for COVID-19 as the main cause of death; and after restricting analyses to statin users who initiated their statin use in the year prior to COVID.

I commend the authors for this interesting study. The large sample size allows for a thorough adjustment for covariates/potential confounders. 

Comments:

1. Abstract - HR is listed as 0.88 - the authors need to specify that this is the adjusted HR.

2. Fig 1 should present the coefficients from the model which adjusts for confounders. 

The main analysis should be presented in the main figure.

Similarity across age groups, sexes, risk groups, etc. should be tested using interaction terms in the main model. The current Fig 1 can be used as a sensitivity analysis

3. The use of the term 'emulated target trial' is very unclear. Either explain this thoroughly, with references, in the methods, or drop the term. Please explain simply what this analysis is and what it shows.

4. Positive control analysis- the authors state that"

"Our positive outcome analysis (benchmark) was consistent with a protective effect of statins (as expected), although the estimated protective effect was greater than that of the benchmark." 

This statement is unclear- Please change to something like:

"In a similar model, the estimated protective effect of statins for non-COVID deaths was greater than for COVID deaths. "

Please state this plainly. Please present the model, in the appendix.

These results needs to be highlighted in the abstract and discussion. It suggests confounding, in that statin users are generally healthier than non-statin users, and this explains the observed protective effect in COVID.

5. There is significant potential for misclassification here - What if only 1 or 2 statin prescriptions were obtained, in early 2019 - this would not be likely to influence outcomes. Why start so early on? I would recommend that the criteria be that a prescription was picked up in the 3 months prior to COVID; or at the very least, perform a sensitivity analysis for those with at least 1 obtained from ~Jan 2020 on? The authors state this could cause additional biases, however the small risk of that does not seem to outweigh the large likelihood of the misclassification with the current methodology.

Reviewer #4: This is an interesting study on the assoication between statin therapy and COVID-19 mortality using the population data from Stockholm. Although well written, there are a few major issues needing attention.

1) The key findings are in Figure 1. We can see the effect of statin changed the direction from increasing the risk (OR 2.02) to protective (OR 0.88) after adjustment of all the confounders. The effect of 0.88 with a p-value of 0.01 is marginal and close to borderline significance. While this still might be possible after adjustment, we really have to scrutinise the statistical analyses to make sure we leave nothing to chance. Also in Figure 1, after adjustment, we can see that over 80 yrs old, IHD, hypertension and heart failure all have protective effect (reduce the risk) of Covid death. Why? This is quite difficult to comprehend and interpret.

2) Competing risk. As the outcome of survival analysis (Cox model) is Covid death rather than all-cause mortality, the competing risk from other deaths needs to be addressed in the analysis. Therefore, survival analysis taking into account of competing risk should be performed as main analysis. As the current result is close to borderline significance and also changes the direction of effect after adjustment, the competing risk issue becomes even more important to make sure the results are robust and reliable.

3) For adjusted confounders, why use age of 45 as a cut-off for inclusion? Is there ethnicity information available? 

4) In results in the abstract and also in the main text, it says "A total of 2 545 individuals died from COVID-19 during follow-up, including 765 (0.5%) statin users and 1780 (0.2%) non-users". However, where are these 0.5% and 0.2% coming from? I can't reproduce these percentages using any figures in Table 1. Can authors please clarity this?

5) In Table 1, for age, median and IQR should be presented in the form of xx (yy-zz) instead.

[LINK]

---

## [Decision Letter · Decision Letter 2]

5 Aug 2021

Dear Dr. Bergqvist,

Thank you very much for submitting your revised manuscript "HMG-CoA reductase inhibitors and COVID-19 mortality in Stockholm, Sweden: a registry-based cohort study" (PMEDICINE-D-21-02232R2) for consideration at PLOS Medicine. 

Your paper was seen again by the reviewers. The reviews are appended at the bottom of this email and any accompanying reviewer attachments can be seen via the link below:

[LINK]

As you will see, one of the reviewers still has some comments that require your attention. We would like to consider a revised version that addresses the reviewers' and editors' comments. Obviously we cannot make any decision about publication until we have seen the revised manuscript and your response, and we plan to seek re-review by one or more of the reviewers. 

We hope to receive your revised manuscript by Aug 26 2021 11:59PM. Please email us (plosmedicine@plos.org) if you have any questions or concerns.

We look forward to receiving your revised manuscript, please let me know if you have any questions. 

Sincerely,

Callam Davidson, 

Associate Editor

PLOS Medicine

plosmedicine.org

Please update the language of your Author Summary as follows:

• Bullet point 1: Lipid-lowering HMG-CoA reductase inhibitors (statins) may influence COVID-19 mortality via their pleiotropic effects on coagulation and the immune system.

• Bullet point 2: Previous studies on this topic have been inconsistent, so we performed a population-based cohort study to investigate the relationship between statins and COVID-19 mortality.

• Bullet point 3: Using data from Swedish health registers, a cohort of 963876 residents of Stockholm, Sweden, were followed from the 1st of March 2020 until the 11th of November 2020. 

Please consider changing the word ‘importantly’ in bullet point 5 of your Author Summary (line 79) to ‘substantially’ or similar. Alternatively, you could say ‘The association was corroborated by sensitivity analyses and was relatively consistent across risk groups’.

For the final bullet point of your Author Summary (line 84), please replace the word ‘total’ with ‘conclusion’ or ‘summary’.

Line 117: Please change ‘set in the rich Swedish health registers’ to ‘using Swedish health register data’, or similar.

Line 117: Please change ‘to determine’ to ‘of determining’.

Line 139: Remove the erroneous hyphen in the word ‘pregnant’.

Line 142: Remove the erroneous hyphen in the word ‘contraindicated’.

Citations should be in square brackets, and preceding punctuation. Please update throughout.

Comments from the reviewers:

Reviewer #2: The authors have responded very well to reviewers' comments

Reviewer #3: The authors have, in my opinion, provided a rigorous and complete rebuttal to my requests and to those of the other reviewers. Although I do not agree with 100% of their answers, I believe they have provided sufficient justification in those few grey areas where some nuanced differences of opinion remain.

I have no further comments at this time.

Reviewer #4: Many thanks authors for their effort to improve the manuscript. Most of my comments were satisfactorily addressed by the authors. However, there is still one remaining issue - the competing risk issue. I am not at all convinced/satisfied with the authors' response on competing risk. To simply censor the other deaths is inadequate and will not solve the competing risk problem in the cause-specific Cox models. To address the competing risk in the Covid death survival analysis, models such as Fine-Gray method are needed as there are potentially quite a few non-Covid deaths in the cohorts which present as competing risk. While it's appreciated that authors gave some references for the arguement, but the vast majority of the stats literature supports the use of the Fine-Gray method (for example) for cause specific survival analysis to effectively and adequately address the competing risk issue. By the way, what is the all-cause mortality in the cohorts? and what's the number of other deaths? what's the proporiton of Covid deaths vs other deaths? So far this competing risk issue hasn't been appropriately addrressed in the paper which could lead to the results and conclusions being subject to scrutiny.

[LINK]

---

## [Decision Letter · Decision Letter 3]

16 Sep 2021

Dear Dr. Bergqvist,

Thank you very much for re-submitting your manuscript "HMG-CoA reductase inhibitors and COVID-19 mortality in Stockholm, Sweden: a registry-based cohort study" (PMEDICINE-D-21-02232R3) for review by PLOS Medicine.

I have discussed the paper with my colleagues and the academic editor and it was also seen again by one reviewer. I am pleased to say that provided the remaining editorial and production issues are dealt with we are planning to accept the paper for publication in the journal.

[LINK]

We look forward to receiving the revised manuscript by Sep 23 2021 11:59PM.   

Sincerely,

Callam Davidson, 

Associate Editor 

PLOS Medicine

plosmedicine.org

Requests from Editors:

Please add the following statement, or similar, to the Methods: "This study is reported as per the Strengthening the Reporting of Observational Studies in Epidemiology (STROBE) guideline (S1 Checklist)."

Line 206: Please include ‘years’ when stating median age of your sample

Table 1: Please define the abbreviations IQR, TIA, and ACE in the footnotes

Figure 1: TIA should be defined as transient ischemic attack (rather than ischemic attack, transient).

Lines 293, 295, 296, 514, 523, 527: Please remove the hyphens from words where they should not appear on these lines.

Comments from Reviewers:

Reviewer #4: Many thanks authors for their great effort to improve the manuscript. The authors have addressed my concerns comprehensively. I am satisfied with the response and revision. No further issues needing attention.

[LINK]

---

## [Editor Report · Decision Letter 4]

20 Sep 2021

Dear Dr Bergqvist, 

On behalf of my colleagues and the Academic Editor, Dr Weiping Jia, I am pleased to inform you that we have agreed to publish your manuscript "HMG-CoA reductase inhibitors and COVID-19 mortality in Stockholm, Sweden: a registry-based cohort study" (PMEDICINE-D-21-02232R4) in PLOS Medicine.

When making the formatting changes, please also make the following update:

* In your data availability statement (part of the submission form), please also include the contact email address (mikrodata@scb.se) as well as the URL you have already provided.

PRESS

Sincerely, 

Callam Davidson 

Associate Editor 

PLOS Medicine